# Auditory-motor synchronization varies among individuals and is critically shaped by acoustic features

Cecilia Mares[1], Ricardo Echavarría Solana[1] & M. Florencia Assaneo [1✉]

The ability to synchronize body movements with quasi-regular auditory stimuli represents a fundamental trait in humans at the core of speech and music. Despite the long trajectory of the study of such ability, little attention has been paid to how acoustic features of the stimuli and individual differences can modulate auditory-motor synchrony. Here, by exploring auditory-motor synchronization abilities across different effectors and types of stimuli, we revealed that this capability is more restricted than previously assumed. While the general population can synchronize to sequences composed of the repetitions of the same acoustic unit, the synchrony in a subgroup of participants is impaired when the unit's identity varies across the sequence. In addition, synchronization in this group can be temporarily restored by being primed by a facilitator stimulus. Auditory-motor integration is stable across effectors, supporting the hypothesis of a central clock mechanism subserving the different articulators but critically shaped by the acoustic features of the stimulus and individual abilities.

[1] Institute of Neurobiology, National Autonomous University of Mexico, Juriquilla, Querétaro, Mexico. ✉email: fassaneo@inb.unam.mx

Auditory-motor synchronization is the ability to coordinate a sequence of motor gestures (i.e., body movements) with a rhythmic auditory stimulus. It is considered an innate human skill[1,2] and a prerequisite for dancing[3], playing an instrument[4] and maintaining a conversation[5]. Furthermore, it has also been proposed to rely on the same neural mechanisms that support time estimations[6], as well as to correlate with different cognitive abilities[7]. Given the ecological validity and many implications of this attribute, its features have been widely investigated in the literature.

The study of auditory-motor synchronization in humans dates back to the early 20th century[8]. The classic task deployed to explore this phenomenon is finger tapping to a metronome, where participants are instructed to tap their finger on a surface along with a rhythmic train of click sounds (or tones). It is by means of this paradigm that the basic features of auditory-motor synchronization have been described. For example (see refs. [9,10] for a complete description), it has been established that: (1) the most efficient rate to synchronize to is around 2 Hz, (2) synchronization is possible only within a range of rates, approximately from 0.5 to 7 Hz[11], (3) the tap precedes the sound by some tens of milliseconds, and (4) there is a fast adaptation to sudden changes in the tempo[12]. Furthermore, biophysical models capable of explaining some of the observed behaviors have been advanced[13,14], and the neural network supporting this ability is relatively well described[15,16].

More recently, this reductionist methodology (i.e., finger tapping) has been complemented by more naturalistic techniques (e.g., using complex stimuli like music and other motor gestures such as walking and clapping[17,18]), which present auditory-motor synchronization as a useful tool to subserve clinical, educational or wellness interventions. For example, auditory-motor entrainment is being used to: develop neurologic music therapies for patients with motor disorders[19], facilitate speech in children with autism[20], and enhance athletic performance[21].

Despite the variety of approaches and many years of research devoted to understanding this fundamental human trait and the potential benefits of auditory-motor training, how individual differences modulate this ability remains poorly explored (at least within the general population). While differences between groups with well-established distinctions have been studied (e.g., adults vs. infants, patients vs. healthy control, professional musicians vs. non-musicians), synchronization is assumed to be attainable by the general adult population with some variation in their performance. In spite of this common assumption, a recent study showed that when the auditory stimulus is a rhythmic train of syllables and the motor gesture comprises the repetition of the syllable "tah", the general population splits into two groups, where only one synchronizes the produced syllables to the perceived syllabic rate[22]. Highlighting the importance of population-level differences, researchers in this first study and a set of follow-up works[22–26] showed that group belonging correlates with structural and functional brain differences, as well as with cognitive abilities.

In the current work, we aimed to establish how this bimodal distribution (i.e., the existence of two groups with qualitatively different behaviors) observed in the speech domain extends (or not) to the general auditory-motor synchronization abilities. We studied a large cohort of participants across different auditory and motor modalities, without assuming unimodal distributions of performance. Results show that auditory-motor synchronization is less general than typically assumed and that individual differences play a crucial role in the description of this phenomenon.

## Results

We evaluated the participants' auditory-motor synchronization abilities across two different types of effectors (hands and vocal tract) and audio stimuli (tones and speech). We instructed them to continuously whisper the syllable "tah" (effector: vocal tract) or to clap (effector: hands) along and in synchrony with a rhythmic auditory stimulus. In all cases, the auditory stimulus comprised a rhythmic sequence (i.e., the repetition of the same tone or a set of syllables) accelerating in time from a presentation rate of 4.3–4.7 units/s (Fig. 1a and "Methods" for more detail). The synchronization values obtained for the four effector-stimulus combinations (i.e., a 4-dimension PLVs vector per subject) were submitted to a random-forest clustering algorithm. Results showed that the optimal model, minimizing the BIC (Bayesian Information Criterion), comprises two clusters (Fig. 1b, $N_{clus} = 2$, $R^2 = 0.603$). A visual exploration of the obtained data (Fig. 1c) suggests that the main difference between the two groups of participants emerges from the speech-like auditory stimulus, irrespective of the effector. To explore in a principled manner the impact that the different effectors and stimuli had on the participants' synchrony, we ran a repeated-measures ANOVA with effector and stimulus type as independent two-level factors. Furthermore, given that the clustering algorithm identified two groups in the population, a high or a low synchronizer (cluster with higher and lower synchronization values, respectively) was included as a between-subjects factor. Results show a significant interaction between the synchrony group and effector (Group*-Eff: $F(1,49) = 6.467$, $p = 0.014$) as well as with stimulus type (Group*Stim: $F(1,49) = 32.254$, $p < 0.001$); no other interaction reached significance (see Supplementary Table 1). Next, we conducted post-hoc analyses to better characterize the significant effects. By averaging across stimuli we found that for high synchronizers, but not for lows, synchronization is facilitated when clapping (Fig. 1d; highs: $t(29) = -4.229$, $p_{Bonf} < 0.001$; lows: $t(20) = -0.222$, $p_{Bonf} = 1.0$). By exploring the effect of the types of stimuli (averaging across effectors), we found that only for low synchronizers tones enhanced synchrony (see Fig. 1e; highs: $t(29) = -2.35$, $p_{Bonf} = 0.14$; lows: $t(20) = -9.371$, $p_{Bonf} < 0.001$).

**Auditory-motor synchronization is impaired for low synchronizers when the audio stimulus comprises speech.** Our first set of analyses showed that synchronization is strongly diminished for the low synchronizers group when the auditory stimulus comprises syllables. Here, we explored to which extent the presence of the auditory stimulus modulates the produced syllabic rate of low synchronizers. In other words, is the synchrony in low synchronizers reduced or impaired? We evaluated a sham synchrony between the sounds produced during the training step (whispering "tah" or clapping) and the auditory stimulus corresponding to the main task (see "Methods"). During the training, participants are instructed to rhythmically produce the corresponding motor gesture while no audio is presented. Thus, the sham synchrony estimates the ability of the participant to internally generate the rhythm without the aid of any external auditory information. For the low synchrony group, we found that when the stimulus comprises speech, the sham synchrony did not differ from the experimental synchronization estimated during the main task (Fig. 2a; $t(20) = 2.092$, $p_{Bonf} = 0.10$). However, synchronization is restored (i.e., synchrony during the main task significantly increases from the sham synchrony) when the acoustic stimulus comprises a train of tones (Fig. 2b; $t(20) = -5.718$, $p_{Bonf} < 0.001$). It is worth noting that the same analysis for the high synchronizers reached significance for both stimulus types (Supplementary Fig. 1a, b; speech: $t(29) = -7.557$, $p_{Bonf} < 0.001$ and tones: $t(29) = -10.641$, $p_{Bonf} < 0.001$).

Additionally, we explore whether the accelerating feature of the auditory stimulus impacts the rate of the low synchrony group. Are they increasing the rate of their speech, even if not matching

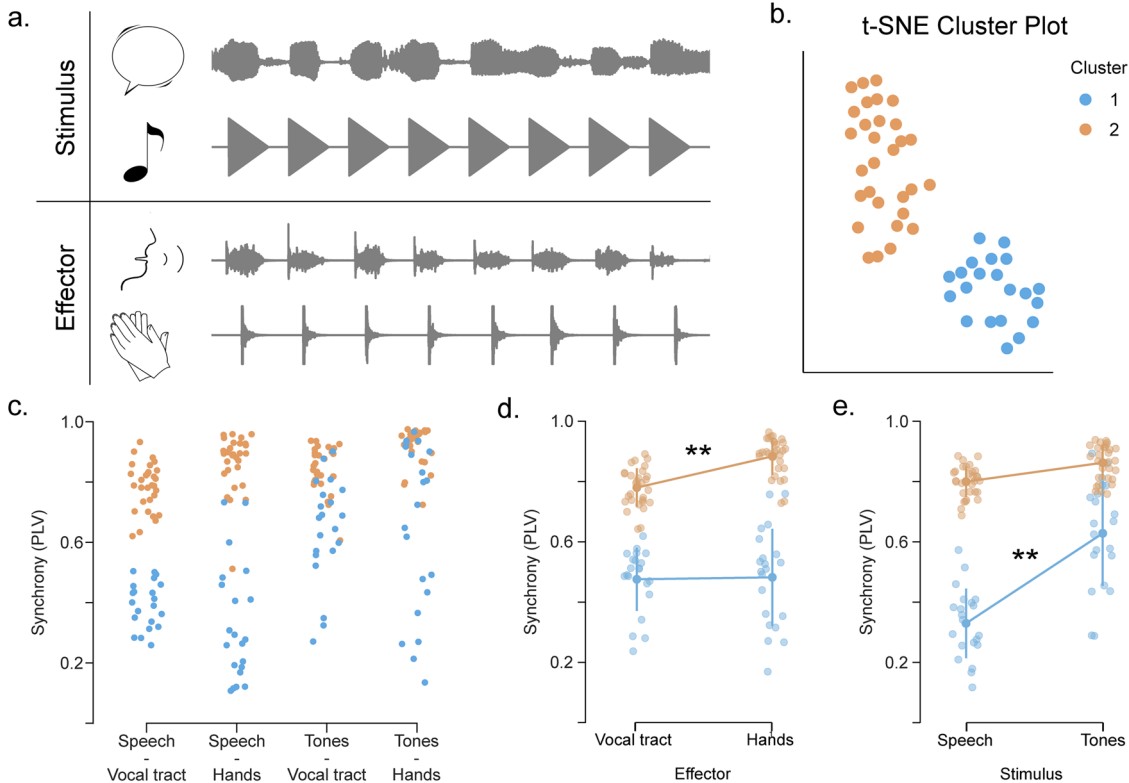

**Fig. 1 Auditory-motor synchronization abilities across different effectors and stimuli. a** Schematic representation of Experiment 1: Participants ($N = 51$) completed four synchronization tasks differing in the effector-stimulus combination. Tasks were completed in separate sessions at least one hour apart from each other. Upper panel: acoustic signals of the different types of stimuli, speech (syllables) and tones. Lower panel: example of the acoustic signals produced by the participants, whispering the syllable "tah" (vocal tract) and clapping (hands). For more detail, see "Methods". **b** t-Distributed Stochastic Neighbor Embedding of the synchronization data. This panel only illustrates the relative distance between the four-dimensional synchronization measurements, axes are uninterpretable. **c** Synchronization across the four different effector-stimulus combinations. In panels (**b**) and (**c**), dots represent individual subjects, and colors are assigned according to the outcome of the random-forest clustering algorithm ($N_{highs} = 30$, $N_{lows} = 21$). **d** Post-hoc comparison averaging across different stimuli. Only high synchronizers improve when clapping. **e** Post-hoc comparison averaging across different effectors. Only low synchronizers improve when listening to tones. In panels (**d**) and (**e**), opaque dots represent mean values, bars SD, shaded dots individual subjects, and **\*\***$p < 0.001$. All panels relate to Experiment 1.

the external frequency? To answer this question, we computed a surrogate synchrony between the sounds produced during the main task and an audio file comprising the same acoustic units as the experimental auditory stimuli but concatenated at a fixed rate of 4.3 Hz. For the low synchrony group, we found the same pattern of results as the one obtained for the sham synchrony. While the experimental synchrony (i.e, the one computed with the accelerated auditory stimulus, the one listened to by the participants) did not differ from the surrogate one for the speech-like stimulus, it surpassed it when the stimulus comprises tones (Fig. 2c, d; speech: t(20) = −8.017, $p_{Bonf} < 0.001$ and tones: t(20) = −11.284, $p_{Bonf} < 0.001$). Again, the same analysis for the high synchronizers reached significance for both stimulus types (Supplementary Fig. 1c, d; speech: t(29) = −35.879, $p_{Bonf} < 0.001$ and tones: t(29) = −41.291, $p_{Bonf} < 0.001$).

These results show that, in a subgroup of the population, auditory-motor integration is impaired for the speech-like stimulus; neither the presence or absence of the stimulus nor its accelerating nature, modify the syllabic rate produced by the low synchronizers.

Next, we explored if, for the auditory stimulus where synchronization is observed (i.e., tones), the low group is actually synchronizing or just reacting to the stimulus. For this goal, we computed the phase lag between the produced sounds (i.e., clapping and "tahs") and the train of tones. The phase lag has been estimated as the phase of the auditory stimulus minus the one of the participant's response (see

"Methods"). Thus, a positive value indicates that the tone precedes the response. We found that when the effector is the vocal tract, participants have phase lag of 15.2° (Fig. 2e) and that this value increases for the hands (Fig. 2f; mean = 54°). Although the phase lag for the hands significantly increased from the one obtained with the vocal tract ($p < 0.001$), synchrony can be assigned for both cases. For a cycle of approximately 222 ms (i.e., presentation rate goes from 4.3 to 4.7 Hz), a phase lag of 54° corresponds to 33.3 ms, a value that is smaller than the typical auditory reaction time (~250 ms[27]). The same phase lag pattern was obtained for the high synchronizers (Supplementary Fig. 1e, f; mean = 3.8° and mean = 29°, respectively; $p < 0.001$).

**Which auditory feature facilitates synchrony in low synchronizers?** For low synchronizers, synchrony was impaired when the auditory stimulus comprised different syllables but was restored when syllables were replaced by a single tone. Two main acoustic differences can be identified between the speech- and the tone-like stimuli: (1) the sharpness of the unit onsets (i.e., at the beginning of each tone the sound amplitude increments from zero to its maximum value, while syllables are coarticulated inducing a soft transition between them, Fig. 1a); and (2) the identity of the repeated unit (i.e., in one case the same tone is being repeated and in the other 16 different syllables are concatenated in random order). Here, we study which of these two acoustic features is facilitating auditory-motor synchrony. For a

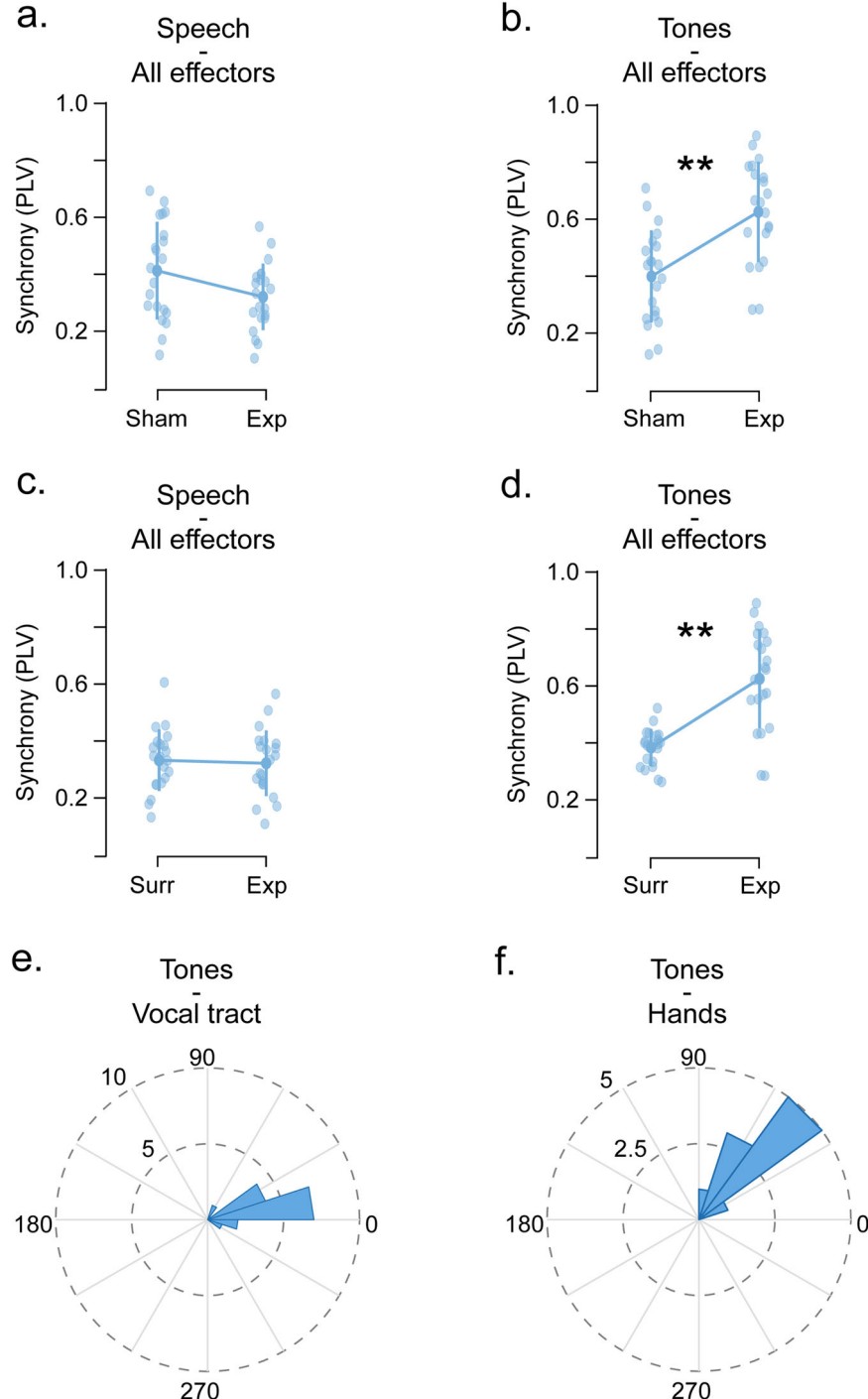

**Fig. 2 Low synchronizers: impaired synchrony to speech is restored for tones. a**, **b** Sham synchrony (i.e., estimated during the training step, where the whispering and clapping are produced without auditory stimulation) compared against the experimental synchrony obtained during the main synchronization task. The average across effectors for speech-like stimulus and tones panel (**a**) and (**b**), respectively ($N = 21$). **c**, **d** Surrogate synchrony (i.e., estimated between the sounds produced during the main task and surrogate audio with a fixed rate of 4.3 Hz) compared against the experimental synchrony (i.e., the one estimated between the sounds produced during the main task and the accelerated stimulus presented to the participants). The average across effectors for speech-like stimulus and tones panel (**c**) and (**d**), respectively ($N = 21$). Opaque dots represent mean values, bars represent SD, shaded dots individual subjects and **$p < 0.001$. **e**, **f** Rose plots depicting the histogram of the mean phase lags between the produced and the perceived sounds. Lag between perceived tones and whispered "tahs" in panel (**e**), hand clapping in panel (**f**). All panels relate to Experiment 1.

group of low synchronizers ($N = 16$), we assessed their degree of synchrony when the auditory stimulus comprised a smooth transition between the same or different tones (Fig. 3a). Then, we explored for each of these conditions if synchrony was increased from the one obtained for the speech-like stimulus. In all cases,

the effector was the vocal tract (i.e., they continuously repeated the syllable "tah"). Results showed that only when the same acoustic unit is repeated synchronization is enhanced (Fig. 3b, c; speech vs. same tone comparison: $t(15) = -4.077$, $p_{Bonf} = 0.002$; speech vs. different tone comparison: $t(15) = 0.418$, $p_{Bonf} = 1$).

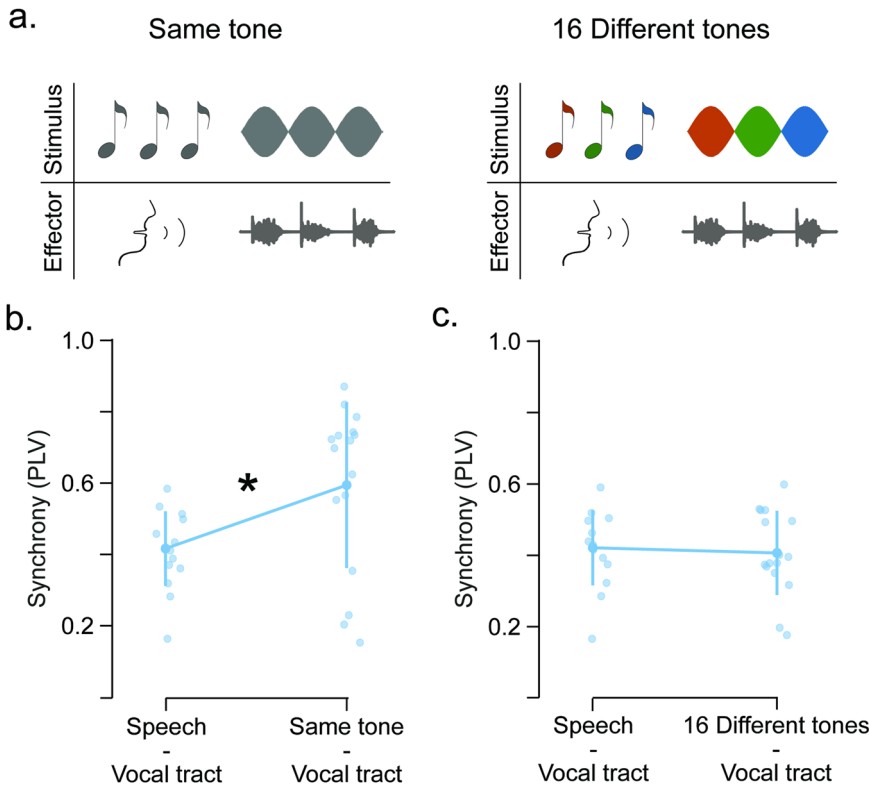

**Fig. 3 The repetition of the same acoustic unit restores synchrony in low synchronizers. a** Schematic representation of Experiment 2. A group of low synchronizers ($N = 16$) completed two extra synchronization tasks, both with the vocal tract as the effector and differing in the acoustic stimulus. One task comprises a sinusoidal modulation of the same tone (i.e., Same tone stimulus). The other task was a sinusoidal modulation of 16 different tones randomly repeated (i.e., 16 Different tones stimulus). **b**, **c** Comparison between the synchronization measurements obtained during Experiment 2 (vocal tract to same tone and vocal tract to different tones, respectively) and the one obtained during Experiment 1 for the vocal tract to speech-like stimulus combination. Opaque dots represent mean values, bars represent SD, shaded dots individual subjects and *$p < 0.05$.

To further test if the repetition of any kind of acoustic unit restores the synchrony in the low synchrony group, we conducted an extra control experiment. Here, a group of low synchronizers ($N = 12$) completed two synchronization tasks with the vocal tract as effector. In one task, the synchronization was assessed for the various syllable stimulus (i.e., the same speech-like stimulus used in Experiment 1, comprising the 16 different syllables). The other task, instead, comprised a repeated speech stimulus where all syllables were replaced by the syllable "go". Results show that synchronization significantly differs between tasks, with synchronization for repeated speech stimulus being higher than for the variable speech one (Supplementary Fig. 2; $t(11) = -4.421$, $p = 0.001$). This outcome demonstrates that synchronization is restored when the same unit is repeated, independently of its particular acoustic features.

**Can rhythmic priming temporarily enhance the abilities of low synchronizers?** Auditory-motor priming has been shown to enhance cognitive[28] and motor abilities[29]. Here we study whether successful auditory-motor synchrony can boost general synchronization abilities. In other words, can the impaired synchronization abilities of the low synchronization group be temporally restored by rhythmic priming? To answer this question, a group of low synchronizers ($N = 15$) completed the vocal tract to speech synchronization task (where synchrony has been shown to be impaired) under different priming conditions. As sketched in Fig. 4a, these conditions were defined by the task executed immediately before: (i) vocal tract to tones synchronization (as described in Fig. 1a), (ii) hands to tones synchronization (also as described in Fig. 1a), (iii) passive listening to the

stimulus of the condition (i/ii), and (iv) no task (i.e., baseline condition). A repeated-measures ANOVA with the condition as an independent factor evidenced that priming has an effect on the low synchronizers' performance (Cond: $F(3,14) = 6.021$, $p = 0.002$). Furthermore, a post-hoc analysis showed that auditory-motor synchronization is restored (significantly increased from baseline) if elicited before by the facilitator stimulus, regardless of the trained effector (Fig. 4b; baseline vs. condition (i): $t(14) = -3.25$, $p_{Bonf} = 0.014$; baseline vs. condition (ii): $t(14) = -3.816$, $p_{Bonf} = 0.003$). However, this carryover effect was not recovered by simply listening to the facilitator stimulus (Fig. 4b; baseline vs. condition (iii): $t(14) = -1.504$, $p_{Bonf} = 0.841$).

In addition, we evaluated a group of high synchronizers ($N = 16$) with condition (ii). This group did not show the carryover effect (Supplementary Fig. 3a, b; baseline vs. condition (ii): $t(15) = -0.979$, $p = 0.343$).

**Does gender and musical training modulate auditory-motor abilities?** The present study focuses on the role played by individual differences in general auditory-motor synchronization abilities. As such, we complemented our results by exploring the effect of the demographic aspects typically associated with this ability: gender and musical background. On one hand, years of musical training have been reported to correlate with auditory-motor synchronization abilities[22,30]. On the other hand, it has been shown that males display faster finger-tapping rates than women[31]. To explore if these variables influence the participants' degree of synchrony, we performed two new analyses for the data collected in Experiment 1 (Fig. 1a). First, we explored if there

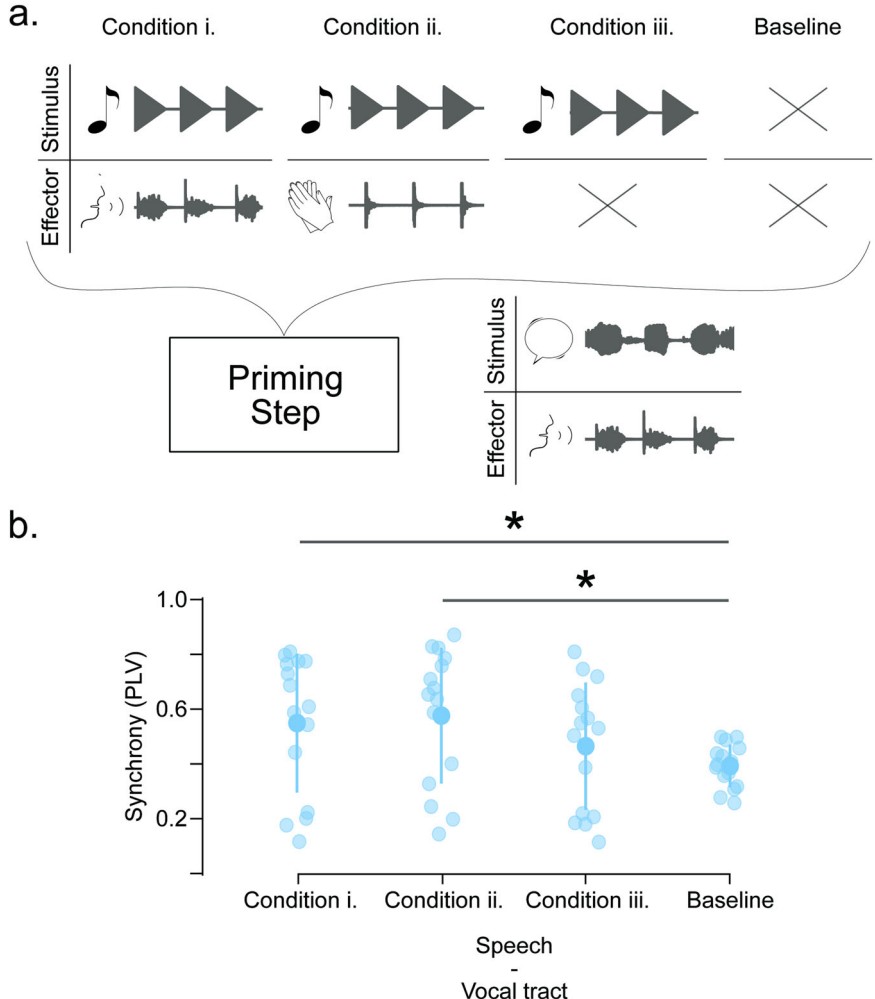

**Fig. 4 Low synchronizer abilities can be enhanced by an active rhythmic priming step. a** Schematic representation of Experiment 3. A group of low synchronizers ($N = 15$) repeated the vocal tract to speech-like stimulus synchronization tasks right after different "priming" conditions. Active priming conditions implied (i) whispering "tah" or (ii) clapping in sync with the train of tones used in Experiment 1. In the passive condition (iii) participants passively listened to the same auditory stimulus. The baseline condition represents the synchronization task performed in Experiment 1 (i.e., vocal tract to speech-like stimulus without any priming). **b** Synchronization measurement obtained from the vocal tract to speech-like stimulus synchronization tasks completed after the different priming conditions. Opaque dots represent mean values, bars represent SD, shaded dots individual subjects and *$p < 0.05$. All panels relate to Experiment 3.

were gender and/or musical training differences between the high and low synchronizer groups. As previously reported[22], groups did not differ in the number of females/males (Supplementary Fig. 4a; lows: $N_{fem}/N_{tot} = 0.57$; highs: $N_{fem}/N_{tot} = 0.50$, $p = 0.77$) and high synchronizers showed more years of musical experience than low synchronizers (Supplementary Fig. 4b; $t(48) = -2.269$, $p = 0.028$). Second, we replicated the repeated-measures ANOVA reported at the beginning of this section, adding gender as an extra independent two-level factor and years of musical training as a covariate. Results showed that all previously reported interactions remained significant (Supplementary Table 2), ensuring that group classification exceeds the explanation power of the participants' musical background. In addition, two other interactions reached significance: Eff*Group*Gender ($F(1,45) = 5.023$, $p = 0.03$) and Stim*Group*Gender ($F(1,45) = 5.41$, $p = 0.025$). To better understand these significant interactions, we conducted further post-hoc analyses. When averaging across stimuli and performing comparisons between genders, we found that males performed better than females only when clapping and for the low synchrony group (Fig. 5a; lows$_{VocalTract}$: $t(19) = 0.038$, $p_{Bonf} = 1.0$; lows$_{Hands}$: $t(19) = -3.333$, $p_{Bonf} = 0.035$; highs$_{VocalTract}$:

$t(27) = -0.771$, $p_{Bonf} = 1.0$; highs$_{Hands}$: $t(27) = -0.036$, $p_{Bonf} = 1.0$). Similarly, when averaging across effectors, the males' outperformance was restricted to the low synchrony group when the stimuli comprised tones (Fig. 5b; lows$_{Speech}$: $t(19) = 0.642$, $p_{Bonf} = 1.0$; lows$_{Tones}$: $t(19) = -3.891$, $p_{Bonf} = 0.005$; highs$_{Speech}$: $t(27) = -0.21$, $p_{Bonf} = 1.0$; highs$_{Tones}$: $t(27) = -0.587$, $p_{Bonf} = 1.0$).

## Discussion

In the current study, we investigated the general auditory-motor synchronization abilities of a cohort of adults without assuming that all participants belong to the same population. By submitting the registered data to an unsupervised clustering algorithm, we identified two different groups in the general population. We named the groups high and low synchronizers, given that one exhibits better performance than the other. Notably, these groups display qualitatively different features. For the low group, synchronization was impaired for the speech-like auditory stimulus and restored when the stimulus was replaced by the repetition of a single tone. Contrary to this behavior, synchronization remained stable across the different stimulus types for the high

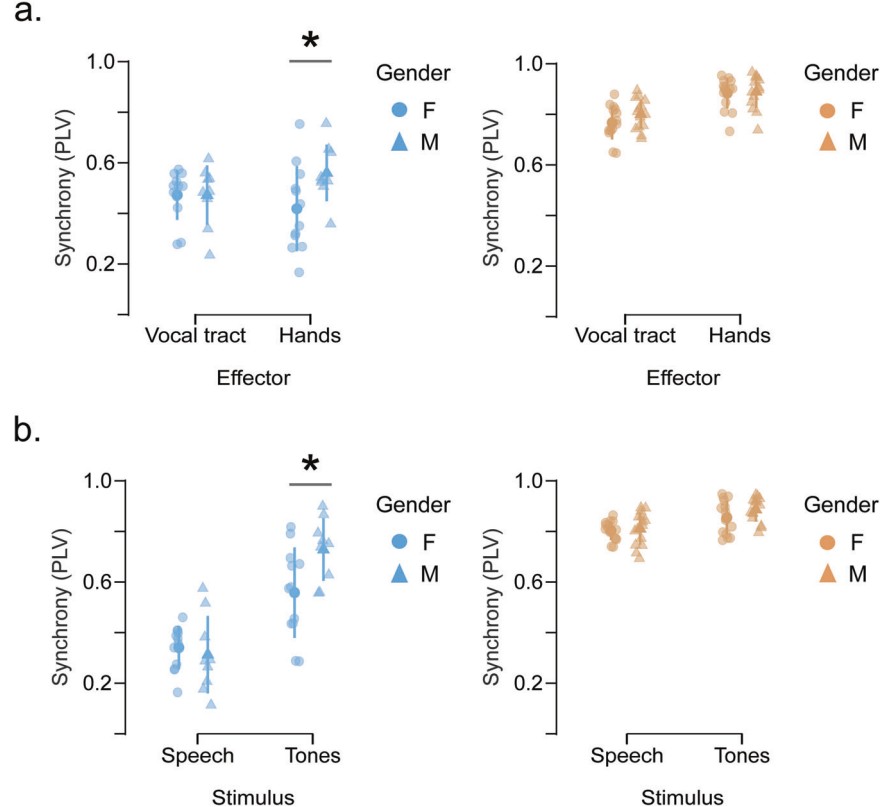

**Fig. 5 Gender differences in synchronization abilities.** Post-hoc comparisons between genders averaging across stimuli and effectors in panels (**a**) and (**b**), respectively ($N_{highs} = 30$, $N_{lows} = 21$). Only low synchronizers display a significant gender difference restricted to the acoustic stimulus comprising tones and clapping. Opaque dots represent mean values, bars SD, shaded dots individual subjects, *$p < 0.05$; triangles and circles indicate males and females, respectively, and orange and blue indicate high and low synchronizers, respectively. All panels relate to Experiment 1.

group. In addition, only the latter showed a preferred effector (i.e., they performed better with the hands than with the vocal tract).

Auditory-motor synchronization has been assumed to be an inherent trait[2,32] of humans and has been widely explored as such (i.e., assuming that any individual can synchronize, with better or worse performance, to any sound). However, our pattern of results evidences that synchronization is more restricted than previously assumed. While it is true that part of the population can easily synchronize to any rhythmic sound, another subgroup requires some regularity in the acoustic features of the units composing the rhythmic sequence to achieve synchrony. Despite the extensive research on auditory-motor synchronization, almost all protocols use metronome-like stimuli consisting of the repetition of the same "click" or tone[9,10]. The literature lacks studies systematically exploring how different acoustic features of the presented stimulus can modulate the degree of auditory-motor synchrony. This can explain why the existence of these two different groups has not been reported before.

In addition, we found that synchronization abilities can be temporarily enhanced by rhythmic priming only for the low synchrony group. Importantly, for the high synchrony group priming does not result in any enhancement, even when improvement was still possible (i.e., synchrony was not a ceiling, given that an enhancement was observed when clapping instead of whispering). These results support the proposal of motor systems behaving as oscillators[33,34] which can be coupled to the auditory system[35,36]. The high synchronizers show an extended range of synchronization (presumably due to strong fronto-temporal white matter connectivity[22]), meaning that they can successfully entrain to external frequencies distant from the internal initial rate of the

motor system. However, for low synchronizers, the internal motor system must be initiated at a rate close enough to the external rate[37] for synchronization to be achievable. Additionally, it is worth noting that the synchronization after the rhythmic priming is highly variable (as shown in Fig. 4b). Implying that not all low synchronizers displayed the enhancement in synchrony induced by the priming. This result suggests that while the motor system can shift its natural rate through rhythmic priming, this modified state is not stable, and it can be easily disturbed. More research will be required to test this hypothesis.

The different phase lags across effectors (i.e., hands having a significantly higher phase lag than the vocal tract) can also be interpreted within the motor oscillator frame. A well-described phenomenon for coupled oscillators is the phase precession occurring when the external rate departs from the internal one[38]. While 4–5 Hz represents reasonable rates for the tongue[39], they may be too fast compared to the natural rhythms of the hands[40]. This could explain why the vocal tract synchronization occurs approximately at zero phase lag while the clapping shows an increased phase lag value. Also, for vocal tract synchronization, the typically reported negative mean synchrony[41] (i.e., the classic observation in the finger tapping protocol that taps precede the auditory stimulus) is present in our data. Although the computed phase lag approximates zero in both groups, this value is estimated using the produced acoustic signal, which is preceded by the vocal tract occlusion. The sound onset time of the "tah" corresponds to the occlusion's release, which happens around 50 ms after the tip of the tongue reaches the palate[42]. In our design, the time point at which the tip of the tongue reaches the palate corresponds to the tap time in the classic finger-tapping protocol, and it precedes the tone by several milliseconds.

Whether synchronization is supported by a general domain or effector-specific "timekeeper"[43] has been discussed in the literature. In line with previous research[44,45], the fact that clapping priming results in an enhancement in vocal tract synchrony (Fig. 4) argues in favor of a general domain motor clock. Existing results pointing in the opposite direction[29] could derive from including high synchronizers in the experimental cohort, thus diluting the priming effect.

Finally, gender differences became noticeable when restricting our analysis to a subgroup of the population and stimulus/effector type. Auditory-motor synchronization reflects not only the cognitive ability to follow a rhythm but also the constraints of the peripheral system[46]. For example, trying to rhythmically jump at 4 Hz would be impossible even for a proficient synchronizer. This would reflect a limitation of the anatomical system (i.e., body inertia, muscle strengths, etc.) and not a cognitive impairment. Based on our results, we hypothesize that the peripheral limitations become relevant when synchronization is unstable. For example, it has been shown that males can tap faster than women[47], presumably due to different anatomical configurations. This could explain why significant differences appeared in the low group when clapping, given that 4 Hz represents a fast rate for hands, but not for the vocal tract, an effector comfortable at this range of rhythms. High synchronizers have enough cognitive resources to compensate for small anatomical imbalances, which is why no significant gender differences were detected. In addition, differences between genders appear for the low group only when the stimulus comprises tones, a condition where synchronization is achievable but clearly diminished if compared with highs (i.e., synchrony is present but in an unstable way).

To summarize, the current work reveals that auditory-motor synchronization considered an inherent human trait and a prerequisite for vocal communication is more restricted than previously assumed and displays substantial individual differences. While synchronization is universally achievable when the auditory stimulus comprises the repetition of the same acoustic unit, it is impaired for a subgroup of the population when stimuli comprehend different units. This outcome highlights the importance of acknowledging individual differences and invites to revise the literature on auditory-motor interaction, given the existence of two groups with qualitatively different behaviors.

## Methods

**Participants**. A first group of 51 participants carried out Experiment 1 (27 females; mean age, 27 years; range, 21–37 years). A second group of 16 participants completed Experiment 2 (10 females; mean age, 27 years; range, 23–40 years). Fourteen participants of this group had already been part of Experiment 1. A third group of 31 participants performed Experiment 3 (14 females; mean age, 26 years; range, 20–37 years); 29 of them had previously participated in Experiment 1. Originally, five extra participants participated in this study but were removed because they did not complete the tasks successfully (i.e., they set the volume too loud such that the stimulus leaked in the recording, the rate of the motor gesture was equal to or less than 2 Hz, or participants remained silent for more than 3 s).

The number of participants was determined based on previous studies with similar protocols. Previous work exploring the auditory-motor synchronization abilities in the general population reported positive results with numbers of participants in the same order of magnitude as in Experiment 1[24,48]. For the two other experiments, the number of participants was smaller since subjects were already classified as high or low synchronizers (Experiment 2: 16 low synchronizers; Experiment 3: 15 low synchronizers and 16 high synchronizers). When dealing with better-characterized populations, positive results have been reported with the number of participants in the range of 10–15[49–51].

All participants were native Mexican Spanish speakers, with self-reported normal hearing and no neurological deficits. All participants answered how many years they had been practicing one or more musical instruments and provided written informed consent. They were paid for taking part in the study. All protocols were approved by the Ethics Committee of the Instituto de Neurobiología of Universidad Nacional Autónoma de México (protocol 096.H).

**Synchronization task**. Throughout the study, participants completed a set of synchronization tasks, varying the stimulus–effector combination. Following Lizacano et al.[52], a synchronization task comprehends the following steps: (1)

Volume adjustment: Participants are asked to set the volume at the maximum level possible, without feeling any discomfort. (2) Training step: Participants passively listen to a 10-s stimulus example consisting of a 4.3 Hz rhythmic sequence; once it ends, they are asked to perform a rhythmic motor gesture (which depends on the current effector), at the same pace as the example for another 10 s; (3) Main task: Participants listen to a rhythmic auditory stimulus (see below) for 50 s and are asked to simultaneously repeat a motor gesture (see below) in synchrony with the external rate. The stimulus and effector varied in each experiment.

**Synchronization stimuli**. All stimuli consisted of a rhythmic sequence of different acoustic units. The rhythmic structure was constant across stimuli: the presentation rate of the acoustic units was progressively increased from 4.3 to 4.7 units/s, using steps of 0.1 units/s; each rate was kept constant for 10 s, so each stimulus lasted 50 s. Four different stimuli were used throughout the study.

Speech, Experiments 1 and 3: Here, the acoustic units are 16 different syllables, made up of two Spanish phonemes, in a consonant-vowel fashion. Syllables were pseudo-randomly concatenated (the same syllable was never repeated consecutively) and the audio file was synthesized using the MBROLA text-to-speech synthesizer[53] with the Spanish Male Voice "es2" at 16 kHz.

Tones, Experiments 1 and 3: A train of 1 kHz tones was synthesized using MATLAB. Each tone was modulated with a triangular function with a fall time of 150 ms, resulting in a pronounced attack.

Various tones, Experiment 2: The acoustic units were 16 tones with logarithmically separated frequencies, ranging from 0.5 to 2 kHz: 0.5, 0.55, 0.6, 0.66, 0.72, 0.79, 0.87, 0.95, 1.05, 1.15, 1.25, 1.38, 1.51, 1.66, 1.82 and 2 kHz. Tones were modulated by a sine function, so they had a soft onset and offset, and the same accelerating features of the previous stimuli were kept. The sine function was set to: $\sin(2\pi f \ t)$, with: f = 4.3 if t < 10, f = 4.4 if 10 < t < 20, f = 4.5 if 20 < t < 30, f = 4.6 if 30 < t < 40 and f = 4.7 if 40 < t. A 50-s-long train of these pseudo-randomly repeated tones (i.e., the same tone was never repeated consecutively) was synthesized using MATLAB.

Same tone, Experiment 2: A train of 1 kHz tones was synthesized using MATLAB. The attack of each tone was modulated by the same sine function as for the Variable tones stimulus.

**Synchronization effectors**. Two different effectors were used throughout this study. Vocal tract: Participants were instructed to continuously whisper the syllable 'tah'. Hands: Participants were asked to continuously clap with both hands.

**Synchronization measurement**. To compute the amount of synchrony between the stimulus and a participant's response, the PLV (Phase-Locking Value) was estimated according to the following formula:

$$PLV = \frac{1}{T} |\sum_{t=1}^{T} e^{i(\theta_1(t) - \theta_2(t))}| \tag{1}$$

where $t$ is the discretized time, $\theta_1$ and $\theta_2$ are the phases of the envelope of the participant's response (i.e., the recordings of the produced sounds, claps or "tahs") and the envelope of acoustic stimulus, respectively, and $T$ is the total number of time points within a 5-s window. Envelopes were resampled at 200 Hz and filtered between 3.3 and 5.7 Hz, and their phases were extracted by means of the Hilbert transform. For each synchronization task, the PLV was estimated on 5-s sliding windows with a 2-s overlap. Finally, to obtain one synchronization value per task, the PLVs were averaged across time windows.

The sham synchrony was estimated as the PLV between the envelope of the sounds produced during the training step (whispers or claps) and the envelope of the first 10 s of the stimuli corresponding to the main task.

The mean phase lag for each participant and synchronization task was computed using the *circ_mean* function from the Circular Statistics Toolbox of MATLAB[54]. More precisely:

$$mean\,phase\,lag = \,circmean(\theta_1(1,2,...,T) - \theta_2(1,2,...,T)) \tag{2}$$

where $\theta_1$ and $\theta_2$ are the phases of the envelopes of the participant's response and the acoustic stimulus, respectively, and $T$ is the discretized total task time.

**Experimental procedures**. All experiments were conducted in a sound-attenuated room, in which participants were seated in front of a computer. All audio stimuli were presented binaurally at a variable sound pressure adjusted by the participant (the maximal volume reached by the used device and stimuli was 100 dB), via ETYMOTIC ER1 headphones, attached to ER1-14A earplugs. Participants' responses were recorded through the internal microphone of the computer.

Experiment 1: Participants were required to complete four synchronization tasks. These were composed of different stimulus–effector combinations, presented in the following order: speech-vocal tract, speech-hands, tones-vocal tract, and tones-hands. Each synchronization task took place in a different session, and the time between sessions was between 1 to 12 h.

Experiment 2: Participants performed two sessions of synchronization tasks, with the following stimulus–effector combinations: various tones-vocal tract and same tone-vocal tract. Between sessions, participants paused between 1 and 8 h.

Experiment 3: Participants accomplished three sessions. In each session, the synchronization task consisting of whispering "tah" concurrently with the speech-like stimulus (stimulus–effector: speech-vocal tract, same as in Experiment 1) was preceded by a different priming task. In the first session, participants were primed by whispering "tah" along and in synchrony with the Tones stimulus. In the second one, the priming task consisted of participants clapping in synchrony with the Tones. Finally, in the third condition, participants were primed by passively listening to the Tones stimulus. The time between sessions was between 1 to 7 h.

**Statistics and reproducibility**. To evaluate the number of clusters that best fit the data of Experiment 1, a random-forest clustering algorithm[55] was used; it provided the number of clusters that best fit the synchronization values from Experiment 1 (Fig. 1b). The interaction of effector and stimulus of the same experiment was determined by a repeated-measures ANOVA with two within-subject factors: effector (vocal tract vs. hands) and stimulus (syllables vs. tones). Group remained as the between-subject factor. Multi-comparison two-tailed Student's $t$-test using a Bonferroni corrected post-hoc analyses were applied to further understand the Group*Eff (Fig. 1d) and Group*Stim (Fig. 1e) interactions.

To assess the differences between the synchrony in the training step and the main task, a two-tailed Student's $t$-test for paired samples was performed (Fig. 2a, b).

For the comparison of the preferred phase of the participant's responses between groups, a Circular non-parametric test from the Circular Statistics Toolbox of MATLAB[54] was used (Fig. 2c, d and Supplementary Fig. 1c, d).

A two-tailed Student's $t$-test for paired samples was used to compare the differences in Experiment 2 (Fig. 3b, c) and data from Experiment 3, comparing results to the baseline condition (Fig. 4b and Supplementary Fig. 3b).

A Fisher's exact test was applied to explore for no nonrandom associations between synchrony group and gender (Supplementary Fig. 4a). A two-tailed Student's $t$-test for independent samples was used to compare the years of musical training between high and low synchronizers (Supplementary Fig. 4b).

Finally, another repeated-measures ANOVA was conducted for Experiment 1 with two within-subject factors: effector (vocal tract vs. hands) and stimulus (syllables vs. tones); two between-subject factors: Group and gender, and years of musical training as covariate. Multi-comparison two-tailed Student's $t$-test using a Bonferroni corrected post-hoc analysis were applied to further understand the Gender*Group*Eff (Fig. 5a) and Gender*Group*Stim (Fig. 5b) interactions.

Data processing and analyses were conducted using custom MATLAB[56] code and Jasp[57].

**Reporting summary**. Further information on research design is available in the Nature Portfolio Reporting Summary linked to this article.

## Data availability

Source data underlying the findings of this study are available in Supplementary Data 1. All other data are available from the corresponding author.

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

## Acknowledgements

We thank Leopoldo González-Santos for providing guidance on computer code development and Jessica González Norris for proofreading the manuscript. This work was supported by UNAM-DGAPA-PAPIIT IA202921/IA200223 and IBRO Return Home Fellowship (M.F.A.). C.M. received CONACYT funding (CVU: 1086447) from the Mexican government.

## Author contributions

M.F.A. conceived and supervised the project. C.M. and R.E.S. collected the data. M.F.A. and C.M. wrote the initial version of the manuscript. All authors analyzed the data and contributed to reviewing process of the manuscript.

## Competing interests
The authors declare no competing interests.
