## [Peer Review File · Communications Biology]

Reviewers' comments:

Reviewer #1 (Remarks to the Author):

The authors study sensorimotor synchronization to auditory stimuli. Experimental manipulations include stimulus type (speech/tones), effector (vocal/clap), and additional analysis factors include synchronization ability (low/high), and gender. They find interesting interaction effects, e.g. tones vs speech improve synchrony but only for low-synchronizers, and hands improve synchrony but only for high-synchronizers.

The manuscript is well-written. The introduction is succinct and clear, the results are well organized and the discussion is thorough. The quality of the figures is very good and have a clear relationship with the results. I have some concerns, though, so I recommend revision before publication.

MAJOR COMMENTS

1. In subsection "Auditory-motor synchronization..." the authors make the claim that synchronization is impaired (instead of just reduced) for low synchronizers with speech stimulus. To show this they take the training data (i.e. free motor gestures, no stimuli--traditionally known as continuation) and pair them with an imaginary constant-period sequence of stimuli to compute a surrogate synchronization value. Then the training step and main task are compared.

1.a. (Minor suggestion) I suggest changing the wording "synchrony" and "synchronization" when referring to the training data, as subjects were not actually synchronizing to anything but in "free clapping/speaking". At least enclose it in quotes with a modifier like "surrogate" or "sham" as in Figure 2.

1.b. (Major concern) The authors conclude that "the presence or absence of the stimulus does not modify the behavior of low synchronizers". I find this conclusion hardly supported by the data. First, the stimuli sequence in the main task is continuously accelerating from 4.3 Hz to 4.7 Hz, do the authors claim that the subjects were not adjusting their motor actions to it? Second, the authors are comparing two different tasks that are most likely supported by different brain regions (Lewis et al 2004, Teghil et al 2019, Repp & Su 2013). Third, in order to show that synchronization is impaired I would suggest comparing it to random (continuation is not), for instance by computing the PLV after shuffling stimuli and responses. I suggest revising this subsection. I think the rest of the manuscript would do just fine if "impaired" is replaced by "reduced/altered".

2. Line 160. Subsection "Which auditory feature..." In Figure 3b the authors compare between speech stimuli and same tone stimuli and conclude that the repetition of the same acoustic unit makes synchronization improve. There is an additional hint that repetition might have an effect by itself: comparison same tone vs 16 different tones. However, two aspects of the stimuli are changed in the comparison in Figure 3b: the repetition (same vs different) and the stimulus type (speech vs tone), so it is difficult to find a causal relationship between a single factor and the effect. Why didn't the authors compare between "speech" and "repeated speech"? (a stimulus of the same syllable repeated 16 times) In fact for low-synchronizers switching from speech to tones has a significant effect (Figure 1e), so this could potentially account for the whole effect shown in Figure 3b. I think you would need to control for it. I'm not asking for any additional experiment, but please discuss.

MINOR COMMENTS

3. What is the purpose of accelerating stimuli? (as opposed to the more traditional constant-period stimulus sequences)

4. Please indicate which experiment you are analyzing in each subsection. For instance, I assume subsection "Which auditory feature..." describes results from experiment 2 based on the number of subjects (N=16), and that subsection "Can rhythmic priming..." describes experiment 3 (but then I found N=15 in the text and N=45 in Methods). Please clarify.

5. Line 84. "The synchronization values obtained for the four effector-stimulus combinations were submitted to a random-forest clustering algorithm". Please clearly state what the main observable "synchronization value" is. I imagine it is the 4D-vector PLV?

6. Line 146. The authors show that the low synchronizers are actually synchronizing instead of reacting. It would be helpful to read a clear interpretation of phase lag sign and behavior (well before reaching the Discussion where there is a paragraph dedicated to this). For instance, in the Introduction the authors note that in finger tapping the taps typically precede the tones by some tens of milliseconds. In this work, does a positive phase lag mean that actions succeed stimuli? (Additionally, in that case, please consider that the 222 ms period plus the observed 33 ms lag would indeed be compatible with a reaction time, i.e. reacting to the stimulus before. Please discuss.)

7. Line 401. The phase lag is an angle. Wasn't the mean phase lag computed via circular statistics?

8. Line 426. Please provide details about the random forest clustering. Was it computed before the t-SNE reduction, or after?

9. Figure 1a. The first triangle is smaller than the others, does it mean anything?

10. Figure 1b. Please indicate axis titles.

11. Figures 2, 3 and 4. please indicate whether the p-values are corrected for multiple comparisons like in Figure 1.

12. Figure 4. Panel b shows that in the baseline condition the subjects are all consistently bad at synchronizing. In the other conditions there seems to be substantial variability across subjects (it is mean +/- SD across subjects, right? It is not stated), as if only some of them can actually make use of the priming. Please discuss.

Reviewer #2 (Remarks to the Author):

The authors investigated individual differences in how acoustic features of stimuli effect in sensorimotor synchronization (SMS) abilities in both clapping and speaking modalities. The results indicate a subpopulation struggles with SMS accuracy to rhythmic spoken syllables and tones, but only when synchronizing with stimuli that vary over a sequence, such as with tones of differing pitches, or different spoken syllables. In addition, differences were seen across genders in the low performing group with males outperforming females when clapping, but not for vocal synchronization, and when synchronizing with tones, but not for synchronizing with vocals. Further findings indicate a cross-effector-modality priming effect, that the authors interpret as evidence of modality general effector timekeeping. The overall results show highlight the importance of factoring in individual differences when researching SMS capabilities.

Overall this paper is well written with findings that should be of interest across multiple sub fields. I

have one suggestion and a number of minor points that will need clarification (mostly in the methods). I suggest the authors may wish to apply an autocorrelation analysis to the effector data. Lag1 autocorrelation can reveal the amount of error correction that occurs during a synchronization task. This may be of use when comparing across groups, and between the training and synchronization conditions where you would expect different lag 1 autocorrelation results. Here is an example:
Iversen, J. R., Patel, A. D., Nicodemus, B., & Emmorey, K. (2015). Synchronization to auditory and visual rhythms in hearing and deaf individuals. *Cognition*, 134, 232-244.

See the rest of my comments below:

Intro / Results

- 1) 130: is this result just for the low synchrony group?
- 2) 134: saying "the presence or absence of the stimulus does not modify the behavior of low synchronizers" may be overstated. For example, there may be differences related to error correction (lag 1 auto-correlation) that do not show up in the mean phase measurements.
- 3) 160: is this experiment 2? Please clearly indicate so it can be tied to the methods.
- 4) 187: This is experiment 3?
- 5) 201: is this a 4th experiment?

Discussion

- 6) 265: Confusing/missing wording. I do not understand this sentence: "However, our pattern of results evidence that synchronization is less ubiquitous than previously assumed" Please clarify.

Methods

- 7) 334: How were the numbers of participants for the different experiments determined?
- 8) 352-353: Was volume level measured, variation may play a role in the results
- 9) 364: The 4.3-4.7 Hz stimulus presentation rate is very fast for any SMS task. How was the presentation rate determined?
- 10) 366: Were the spoken stimuli in a male voice/female voice? Was the voice always the same?
- 11) 370: what are the duration, and the rise and fall times of the tones?
- 12) 370-378: How was the tone frequency range determined? Were they based on previous works?
- 13) 374: Please describe the sine function to facilitate replication.
- 14) 382: How was it determined to use the 'tah' effector, as opposed to 'dah' or 'bah', for example?
- 15) 364: So, seven total runs for each stimuli? Please clearly indicate the total number of runs for each condition/stimulus type.
- 16) 408: it is stated the stimuli are presented at a mean of 70 db, yet earlier it is stated that subjects adjusted volume? Please clarify
- 17) How were effector onsets determined? By algorithm? By hand? What software was used? Related, how was alignment determined between stimuli timing & effector timing?

Reviewer #1

The authors study sensorimotor synchronization to auditory stimuli. Experimental manipulations include stimulus type (speech/tones), effector (vocal/clap), and additional analysis factors include synchronization ability (low/high), and gender. They find interesting interaction effects, e.g. tones vs speech improve synchrony but only for low-synchronizers, and hands improve synchrony but only for high-synchronizers. The manuscript is well-written. The introduction is succinct and clear, the results are well organized and the discussion is thorough. The quality of the figures is very good and have a clear relationship with the results. I have some concerns, though, so I recommend revision before publication.

We thank the reviewer for the positive overall assessment of our work. We integrated all his/her suggestions into this new version of the manuscript. Please, find below a point-by-point answer to each of the raised comments. Here, as well as in the new version of the manuscript uploaded to the system, you can find all the modifications done highlighted in red.

MAJOR COMMENTS

1. In subsection "Auditory-motor synchronization..." the authors make the claim that synchronization is impaired (instead of just reduced) for low synchronizers with speech stimulus. To show this they take the training data (i.e. free motor gestures, no stimuli-traditionally known as continuation) and pair them with an imaginary constant-period sequence of stimuli to compute a surrogate synchronization value. Then the training step and main task are compared.

1.a. (Minor suggestion) I suggest changing the wording "synchrony" and "synchronization" when referring to the training data, as subjects were not actually synchronizing to anything but in "free clapping/speaking". At least enclose it in quotes with a modifier like "surrogate" or "sham" as in Figure 2.

1.b. (Major concern) The authors conclude that "the presence or absence of the stimulus does not modify the behavior of low synchronizers". I find this conclusion hardly supported by the data. First, the stimuli sequence in the main task is continuously accelerating from 4.3 Hz to 4.7 Hz, do the authors claim that the subjects were not adjusting their motor actions to it? Second, the authors are comparing two different tasks that are most likely supported by different brain regions (Lewis et al 2004, Teghil et al 2019, Repp & Su 2013). Third, in order to show that synchronization is impaired I would suggest comparing it to random (continuation is not), for instance by computing the PLV after shuffling stimuli and responses. I suggest revising this subsection. I think the rest of the manuscript would do just fine if "impaired" is replaced by "reduced/altered".

We thank the Reviewer for this thoughtful comment. Following his/her advice we: changed the word synchrony when referring to the training data to sham

synchrony and we added an extra analysis using a surrogate audio. We believe that this extra analysis strengthens our results by addressing the Reviewers' concerns ("First, the stimuli sequence in the main task is continuously accelerating from 4.3 Hz to 4.7 Hz, do the authors claim that the subjects were not adjusting their motor actions to it? Second, the authors are comparing two different tasks that are most likely supported by different brain regions"). Following the Reviewer's advice, we estimated a surrogate synchrony by pairing the sounds produced during the main task with an audio keeping the same acoustic features as the used stimulus but with a fixed rate of 4.3Hz. Results show that for the low synchronizers, when the stimulus comprises speech, the surrogate synchrony did not differ from the experimental one. This result proves that the low synchronizers do not adjust their motor actions to match the accelerating stimulus. This new analysis has been included into the results section and two extra panels were added to Figure 2 and Figure S1.

Section Auditory-motor synchronization is impaired for low synchronizers when the audio stimulus comprises speech now reads:

"Our first set of analyses showed that synchronization is strongly diminished for the low synchronizers group when the auditory stimulus comprises syllables. Here, we explored to which extent the presence of the auditory stimulus modulates the produced syllabic rate of low synchronizers. In other words, is the synchrony in low synchronizers reduced or impaired? We evaluated a "sham synchrony" between the sounds produced during the training step (whispering "tah" or clapping) and the auditory stimulus corresponding to the main task (see Methods). During the training, participants are instructed to rhythmically produce the corresponding motor gesture while no audio is presented. Thus, the sham synchrony estimates the ability of the participant to internally generate the rhythm without the aid of any external auditory information. For the low synchrony group, we found that, when the stimulus comprises speech, the sham synchrony did not differ from the experimental synchronization estimated during the main task (Fig. 2a; $t(20)=2.092$, $p_{\text{Bonf}}=0.10$). However, synchronization is restored (i.e., synchrony during the main task significantly increases from the sham synchrony) when the acoustic stimulus comprises a train of tones (Fig. 2b; $t(20)=-5.718$, $p_{\text{Bonf}}<0.001$). It is worth noting that the same analysis for the high synchronizers reached significance for both stimulus types (Fig. S1a&b; speech: $t(29)=-7.557$, $p_{\text{Bonf}}<0.001$ & tones: $t(29)=-10.641$, $p_{\text{Bonf}}<0.001$).

Additionally, we explore whether the accelerating feature of the auditory stimulus impacts the rate of the low synchrony group. Are they increasing the rate of their speech, even if not matching the external frequency? To answer this question, we computed a "surrogate synchrony" between the sounds produced during the main task and an audio file comprising the same acoustic units as the experimental auditory stimuli but concatenated at a fixed rate of 4.3 Hz. For the low synchrony group, we found the same pattern of results as the one obtained for the sham synchrony. While the experimental synchrony (i.e, the one computed with the auditory stimulus accelerating from 4.3 to 4.7 Hz, the one listened by the participants) did not differ from the surrogate one for the speech-like stimulus, it surpassed

it when the stimulus comprises tones (Fig. 2c&d; speech: $t(20)=-8.017$, $p_{\text{Bonf}}<0.001$ & tones: $t(20)=-11.284$, $p_{\text{Bonf}}<0.001$). Again, the same analysis for the high synchronizers reached significance for both stimulus types (Fig. S1c&d; speech: $t(29)=-35.879$, $p_{\text{Bonf}}<0.001$ & tones: $t(29)=-41.291$, $p_{\text{Bonf}}<0.001$).

These results show that, in a subgroup of the population, auditory-motor integration is impaired for the speech-like stimulus; neither the presence or absence of the stimulus, nor its accelerating nature, modify the syllabic rate produced by the low synchronizers.”

Figure 2; Low synchronizers: Impaired synchrony to speech is restored for tones. a&b. Sham synchrony (i.e., estimated during the training step, where the whispering and clapping are produced without auditory stimulation) compared against the experimental synchrony obtained during the main synchronization task. Average across effectors for speech like-stimulus and tones panel a and b, respectively (N=21). c&d. Surrogate synchrony (i.e., estimated between the sounds produced during the main task and a surrogate audio with a fixed rate of 4.3Hz) compared against the experimental synchrony (i.e., the one estimated between the sounds produced during the main task and the accelerated stimulus presented to the participants). Average across effectors for speech like-

stimulus and tones, panel c and d, respectively (N=21). Dots represent mean values; bars represent SD and $p<0.001$. e&f. Rose plots depicting the histogram of the mean phase lags between the produced and the perceived sounds. Lag between perceived tones and whispered “tahs” in panel e, hand clapping in panel f. All panels relate to Experiment 1.**

Supplementary Figure 1; High synchronizers’ synchrony remains stable across stimuli. a&b. Sham synchrony (i.e., estimated with the whispering and clapping produced without auditory stimulation) compared to the one obtained during the main synchronization task. Average across effectors for speech like-stimulus and tones panel a and b, respectively (N=30). c&d. Surrogate synchrony (i.e., estimated between the sounds produced during the main task and a surrogate audio with a fixed rate of 4.3Hz) compared against the experimental synchrony (i.e., the one estimated between the sounds produced during the main task and the accelerated stimulus presented to the participants). Average across effectors for speech like-stimulus and tones panel c and d, respectively (N=30). Dots represent mean values, bars SD and $p<0.001$. e&f. Rose plots illustrate the histogram of the mean phase lag between the produced and the perceived sounds. Lag between perceived tones and: whispered “tahs” in panel e, hands clapping in panel f. All panels relate to Experiment 1.**

2. Line 160. Subsection "Which auditory feature..." In Figure 3b the authors compare between speech stimuli and same tone stimuli and conclude that the repetition of the same acoustic unit makes synchronization improve. There is an additional hint that repetition might have an effect by itself: comparison same tone vs 16 different tones. However, two aspects of the stimuli are changed in the comparison in Figure 3b: the repetition (same vs different) and the stimulus type (speech vs tone), so it is difficult to

find a causal relationship between a single factor and the effect. Why didn't the authors compare between "speech" and "repeated speech"? (a stimulus of the same syllable repeated 16 times) In fact for low-synchronizers switching from speech to tones has a significant effect (Figure 1e), so this could potentially account for the whole effect shown in Figure 3b. I think you would need to control for it. I'm not asking for any additional experiment, but please discuss.

Reviewer is right. In order to strengthen our results, we conducted the suggested control, and we included it as Supplementary material. This new version of the manuscript includes the following description in the main text and the corresponding extra figure in the Supplementary Material section:

Main text:

“To further test if the repetition of any kind of acoustic unit restores the synchrony in the low synchrony group, we conducted an extra control experiment. Here, a group of low synchronizers (N=12) completed two synchronization tasks with the vocal tract as effector. In one task, the synchronization was assessed for the “various syllable” stimulus (i.e., the same speech-like stimulus used in experiment 1, comprising the 16 different syllables). The other task, instead, comprised a “repeated speech” stimulus where all syllables were replaced by the syllable “go”. Results show that synchronization significantly differs between tasks, with synchronization for “repeated speech” stimulus being higher than for the “variable speech” one (Fig. S2 $t(11)=-4.421$, $p=0.001$). This outcome demonstrates that synchronization is restored when the same unit is repeated, independently of its particular acoustic features.”

Supplementary:

Supplementary Figure 2; The repetition of the same acoustic unit restores synchrony in low synchronizers. A group of low synchronizers (N=13) completed two extra synchronization tasks, both

*with the vocal tract as effector and differing in the acoustic stimulus. One task (Variable speech), used the same acoustic stimulus as experiment 1, which comprises a random concatenation of 16 different syllables. In the other task (Repeated speech), all syllables of the stimulus were replaced by the syllable “go”. The plot displays the comparison between the synchronization values obtained for each task. Dots represent mean values, bars SD and * $p < 0.05$.*

MINOR COMMENTS

3. What is the purpose of accelerating stimuli? (as opposed to the more traditional constant-period stimulus sequences)

The present study builds up on previous works of our team showing a bimodal distribution in the general population while assessing the speech-to-speech synchrony using two versions of the same protocol: the Implicit Fixed Version and the Explicit Accelerated one (Lizcano-Cortez et al., 2022). While the first version comprises a constant-period stimulus sequence, as suggested by the Reviewer, it has the disadvantage of the instructions being orthogonal to the synchronization task (i.e., participants are instructed to recall the perceived syllables no explicit instruction is given about the expected syllabic rate). We got previous concerns about different subjects understanding differently the instructions, and this factor being the one leading to the bimodal outcome (despite this explanation is hard to be reconciled with the brain structural differences between the groups). For this reason, we chose the Explicit Accelerated version, which also has been shown to grant a bimodal, and there is no plausible confusion in the instructions interpretation since the experimenter directly asks participants to synchronize to the stimulus. Additionally, the stimulus comprising the sequence of tones was not compatible with the Explicit Fixed Version (i.e., it was not plausible to instruct the subject to recall the auditory units being repeated since it was always the same tone).

4. Please indicate which experiment you are analyzing in each subsection. For instance, I assume subsection "Which auditory feature..." describes results from experiment 2 based on the number of subjects (N=16), and that subsection "Can rhythmic priming..." describes experiment 3 (but then I found N=15 in the text and N=45 in Methods). Please clarify.

We apologize for the lack of clarity and for a mistake committed in the descriptions of the groups. On one hand, the Reviewer is right and there is a mismatch between the N reported in Results and in Methods. There were 31 participants who completed Experiment 3 (16 high and 15 low synchronizers), not 45. We amended this in this new version of Manuscript. Additionally, to clarify which analysis was performed on each experiment we specify this information in the caption of each figure. At the end of each caption, we added the following sentence: “All panels relate to Experiment n”

5. Line 84. "The synchronization values obtained for the four effector-stimulus combinations were submitted to a random-forest clustering algorithm". Please clearly state what the main observable "synchronization value" is. I imagine it is the 4D-vector PLV?

We clarified this in this new version of the manuscript.

"The synchronization values obtained for the four effector-stimulus combinations (i.e., a 4 dimensions PLVs vector per subject) were submitted to a random-forest clustering algorithm."

6. Line 146. The authors show that the low synchronizers are actually synchronizing instead of reacting. It would be helpful to read a clear interpretation of phase lag sign and behavior (well before reaching the Discussion where there is a paragraph dedicated to this). For instance, in the Introduction the authors note that in finger tapping the taps typically precede the tones by some tens of milliseconds. In this work, does a positive phase lag mean that actions succeed stimuli? (Additionally, in that case, please consider that the 222 ms period plus the observed 33 ms lag would indeed be compatible with a reaction time, i.e. reacting to the stimulus before. Please discuss.)

Following the Reviewers advice, we included the sentence highlighted in red in the Results section:

"For this goal, we computed the phase lag between the produced sounds (i.e., clapping and "taps") and the train of tones. The phase lag has been estimated as the phase of the auditory stimulus minus the one of the participant's response (see Methods). Thus, a positive value indicates that the tone precedes the response. We found that when the effector is the vocal tract, participants have phase lag of 15.2° (Fig. 2c) and that this value increases for the hands (Fig. 2d; mean=54°)."

We believe that the misunderstanding with the lag sign led to the confusion regarding adding one period. According to the method used to estimate phase, it is not necessary to add one period. To clarify this point, we computed the phase lag between two short exemplar signals (see Figure R1). We got a mean phase lag of 52.4°, since the rate of the signal is 4.4 Hz it corresponds to approximately 33 msec, a value that aligns well with the observed lag between signals.

Figure R1, Exemplar signals used to estimate the phase lag. In gray an example of a subject's clapping signal and in magenta the sequence of perceived tones. $\Delta t \sim 33\text{ms}$

7. Line 401. The phase lag is an angle. Wasn't the mean phase lag computed via circular statistics?

Reviewer is right, we apologize for the inaccuracy. This new version of the manuscript reads:

“The mean phase lag for each participant and synchronization task was computed using the `circ_mean` function from the Circular Statistics Toolbox of MATLAB⁵¹. More precisely, $\text{mean phase lag} = \text{circmean}(\theta_1(1, 2, \dots, T) - \theta_2(1, 2, \dots, T))$, where θ_1 and θ_2 are the phases of the envelopes of the participant's response and the acoustic stimulus, respectively, and T is the discretized total task time.”

8. Line 426. Please provide details about the random forest clustering. Was it computed before the t-SNE reduction, or after?

10. Figure 1b. Please indicate axis titles.

The random forest clustering was computed on the PLV values, the t-SNE is only for visualization purposes and as stated in JASP the axis are uninterpretable. According to the JASP manual:

t-SNE cluster plot: Generates a t-SNE plot of the clustering output. t-SNE plots are used for visualizing high-dimensional data in a low-dimensional space of two dimensions aiming to illustrate the relative distances between data observations. The t-SNE two-dimensional space makes the axes uninterpretable. A t-SNE plot seeks to give an impression of the relative distances between observations and clusters. To recreate the same t-SNE plot across several clustering analyses you can set their seed to the same value, as the t-SNE algorithm uses random starting values.

To avoid confusions, the main text now reads:

“The synchronization values obtained for the four effector-stimulus combinations (i.e., a 4 dimensions PLVs vector per subject) were submitted to a random-forest clustering algorithm.”

And the Figure's caption states:

“b. t-Distributed Stochastic Neighbor Embedding of the synchronization data. This panel is only for visualization purposes, it illustrates the relative distance between the four-dimensional synchronization measurements, axis are uninterpretable”

9. Figure 1a. The first triangle is smaller than the others, does it mean anything?

The triangle is smaller because we cut the audio signal at a random time and the tone started some milliseconds before. To avoid confusions, we modified the figure choosing a starting time point corresponding to a silence in the sequence of tones. See new figure 1.

11. Figures 2, 3 and 4. please indicate whether the p-values are corrected for multiple comparisons like in Figure 1.

In this new version we reported all the corrected p values and explicitly stated so.

12. Figure 4. Panel b shows that in the baseline condition the subjects are all consistently bad at synchronizing. In the other conditions there seems to be substantial variability across subjects (it is mean +/- SD across subjects, right? It is not stated), as if only some of them can actually make use of the priming. Please discuss.

Reviewer is right, we added the red sentence in the Discussion:

*“The high synchronizers show an extended range of synchronization (presumably due to strong fronto-temporal white matter connectivity²²), meaning that they can successfully entrain to external frequencies distant to the internal initial rate of the motor system. However, for low synchronizers the internal motor system has to be initiated at a rate close enough to the external rate³⁷ in order for synchronization to be achievable. **Additionally, it is worth noting that the synchronization after the rhythmic priming is highly variable (as shown in Fig. 4b). Implying that not all low synchronizers displayed the enhancement in synchrony induced by the priming. This result suggests that while the motor system is able to shift its natural rate through the rhythmic priming, this modified state is not stable, and it can be easily disturbed. More research will be required to test this hypothesis.”***

Reviewer #2

The authors investigated individual differences in how acoustic features of stimuli effect in sensorimotor synchronization (SMS) abilities in both clapping and speaking modalities. The results indicate a subpopulation struggles with SMS accuracy to rhythmic spoken syllables and tones, but only when synchronizing with stimuli that vary over a sequence, such as with tones of differing pitches, or different spoken syllables. In addition, differences were seen across genders in the low performing group with males outperforming females when clapping, but not for vocal synchronization, and when synchronizing with tones, but not for synchronizing with vocals. Further findings indicate a cross-effector-modality priming effect, that the authors interpret as evidence of modality general effector timekeeping. The overall results highlight the importance of factoring in individual differences when researching SMS capabilities. Overall this paper is well written with findings that should be of interest across multiple sub fields.

We very much thank the Reviewer for his/her supportive statements. We modified our Manuscript according to his/her comments, which we believe strongly increased the quality of the study. Please, find below an answer for each of the raised concerns. Here, as well as in the new version of the manuscript uploaded to the system, you can find all the modifications done highlighted in red.

I have one suggestion and a number of minor points that will need clarification (mostly in the methods). I suggest the authors may wish to apply an autocorrelation analysis to the effector data. Lag1 autocorrelation can reveal the amount of error correction that occurs during a synchronization task. This may be of use when comparing across groups, and between the training and synchronization conditions where you would expect different lag 1 autocorrelation results. Here is an example: Iversen, J. R., Patel, A. D., Nicodemus, B., & Emmorey, K. (2015). Synchronization to auditory and visual rhythms in hearing and deaf individuals. *Cognition*, 134, 232-244.

While we acknowledge that lag1, as well as higher-lags autocorrelations, represent informative measures about the sensorimotor synchronization processes, this kind of analysis does not fit well with our experimental design. Our main goal was to assess differences between participants in their degree of synchrony and how those differences are modulated by different effector-stimulus combinations. We aimed to better understand previous work identifying a two-group segregation in the speech-to-speech synchronization abilities of the general population, predicting cognitive skills as well as functional and brain features (Assaneo et al., 2019). A deeper exploration of the specific features of how the synchrony is established was out of the scope of this study. For that reason, we designed a paradigm in which the sounding participants' responses were recorded in parallel with the perceived sounds,

and the synchrony was estimated as the PLV between the envelopes of these two acoustic signals. The envelopes filtered between 3.3 and 5.7 Hz, +/- 1Hz around the syllabic rate, that starts at 4.3 and ends at 4.7 (see Figure 1). We did not compute each response onset; instead, we estimated the synchrony between the two continuous time signals (Lizcano-Cortés et al., 2022). In the present dataset, while it would be relatively easy to define onsets for the clapping and tones, it is not so straightforward for the speech audios (see Figure 1). This work represents a first attempt to characterize individual differences in the degree of synchrony across effectors and stimulus and is successful in the sense that it identifies which acoustic features altered synchrony in a subgroup of the population. We believe that assessing lag1, lag2 and lag3 autocorrelations could represent the goal of a follow up study specifically designed for this goal. The speech stimulus can be constructed by concatenating consonant-vowel syllables (instead of being coarticulated as in this case) starting with stop-unvoiced consonant to get a clear auditory onset. Participants' responses can be collected using electropalatography (<https://icspeech.com/electropalatography.html>) to get the precise time point in which the tongue touches the palate.

Figure R2; Audio files examples. Both panels show acoustic signals in gray; and over imposed in magenta its corresponding envelope filtered between 3.3 and 5.7 Hz. Upper panel: Participant's response when instructed to continuously whisper "tah". Lower panel: Speech-like auditory stimulus.

Assaneo, M. F., Ripollés, P., Orpella, J., Lin, W. M., de Diego-Balaguer, R., & Poeppel, D. (2019). Spontaneous synchronization to speech reveals neural mechanisms facilitating language learning. *Nature Neuroscience*.

Lizcano-Cortés, F., Gómez-Varela, I., Mares, C., Wallisch, P., Orpella, J., Poeppel, D., Ripollés, P., & Assaneo, M. F. (2022). Speech-to-Speech Synchronization protocol to classify human participants as high or low auditory-motor synchronizers. *STAR Protocols*.

See the rest of my comments below:

Intro / Results

1) 130: is this result just for the low synchrony group?

Reviewer is right, we clarified this in this new version of the manuscript. Line 104 now reads:

“For the low synchrony group, we found that ...”

2) 134: saying “the presence or absence of the stimulus does not modify the behavior of low synchronizers” may be overstated. For example, there may be differences related to error correction (lag 1 auto-correlation) that do not show up in the mean phase measurements.

In line with the Reviewer’s observation, we modified this sentence to:

“These results show that, in a subgroup of the population, auditory-motor integration is impaired for the speech-like stimulus; neither the presence or absence of the stimulus, nor its accelerating nature, modify the syllabic rate produced by the low synchronizers.”

3) 160: is this experiment 2? Please clearly indicate so it can be tied to the methods.

4) 187: This is experiment 3?

5) 201: is this a 4th experiment?

We apologize for the lack of clarity. In this new version of the manuscript, for each analysis we clearly stated to which experiment it belongs. At the end of each figure caption, we added the following sentence: *“All panels relate to Experiment n”*.

Discussion

6) 265: Confusing/missing wording. I do not understand this sentence: “However, our pattern of results evidence that synchronization is less ubiquitous than previously assumed” Please clarify.

We hope this modified version is clearer, we would be happy to clarify further if Reviewer considers it necessarily:

“However, our pattern of results evidence that synchronization is more restricted than previously assumed.”

Methods

7) 334: How were the numbers of participants for the different experiments determined?

We aimed to collect data from 55 participants for experiment 1, 15 for experiment 2 and 30 for experiment 3. The number of participants were determined based on previous studies with similar protocols. Previous work exploring the

auditory-motor synchronization abilities in the general population reported positive results with numbers of participants in the same order of magnitude as in experiment 1. For example, N=44 (Versaci & Laje, 2021) and N=55 (Kern et al., 2021). For the 2 other Experiments the number of participants was smaller since subjects were already classified as high or low synchronizers (experiment 2: 16 low synchronizers, we aimed for 15 but an extra participant enrolled, and we kept it; experiment 3: 15 low synchronizers, and 16 high synchronizers, as before, we aimed for 15 but an extra one enrolled). When dealing with better characterized populations positive results have been reported with number of participants in the range of 10 to 15. For example, N=12 musicians and N=17 visual experts (Hove et al., 2013), N=9 pianist (Keller et al., 2007), and N=16 musicians N=16 non musicians (Franěk et al., 1991).

Franěk, M., Mates, J., Radil, T., Beck, K., & Pöppel, E. (1991). Finger tapping in musicians and nonmusicians. International Journal of Psychophysiology

Hove, M. J., Iversen, J. R., Zhang, A., & Repp, B. H. (2013). Synchronization with competing visual and auditory rhythms: Bouncing ball meets metronome. Psychological Research.

Keller, P. E., Knoblich, G., & Repp, B. H. (2007). Pianists duet better when they play with themselves: On the possible role of action simulation in synchronization. Consciousness and Cognition.

Kern, P., Assaneo, M. F., Endres, D., Poeppel, D., & Rimmele, J. M. (2021). Preferred auditory temporal processing regimes and auditory-motor synchronization. Psychonomic Bulletin & Review.

Versaci, L., & Laje, R. (2021). Time-oriented attention improves accuracy in a paced finger-tapping task. European Journal of Neuroscience.

To clarify this procedure, we included the following paragraph in the Participants section:

“The number of participants were determined based on previous studies with similar protocols. Previous work exploring the auditory-motor synchronization abilities in the general population reported positive results with numbers of participants in the same order of magnitude as in experiment 1^{1,2}. For the 2 other experiments the number of participants was smaller since subjects were already classified as high or low synchronizers (experiment 2: 16 low synchronizers; experiment 3: 15 low synchronizers and 16 high synchronizers). When dealing with better characterized populations positive results have been reported with number of participants in the range of 10 to 15³⁻⁵.”

8) 352-353: Was volume level measured, variation may play a role in the results

Volume level has not been measured in this study, because in a previous work we showed that it plays no role in the high vs low synchronizers classification. See: *Lizcano-Cortés, F., Gómez-Varela, I., Mares, C., Wallisch, P., Orpella, J., Poeppel, D., Ripollés, P., & Assaneo, M. F. (2022). Speech-to-Speech Synchronization protocol to classify human participants as high or low auditory-motor synchronizers. STAR Protocols.*

9) 364: The 4.3-4.7 Hz stimulus presentation rate is very fast for any SMS task. How was the presentation rate determined?

The present study builds on previous works of our team showing a bimodal distribution in the general population while assessing the speech-to-speech synchrony using a behavioral test with the accelerating properties pointed by the Reviewer (Lizcano-Cortez et al., 2022). Crucially, group belonging correlates with structural and functional brain differences, as well as with cognitive abilities (Assaneo et al., 2021, 2022; Assaneo, Rimmele, et al., 2019; Assaneo, Ripollés, et al., 2019; Orpella et al., 2022), highlighting the suitability of the test to identify population-level differences. The goal of the current work was to identify which was the feature of the behavioral test allowing to identify the bimodal distribution in the general population, as well as to further describe the auditory-motor synchronization differences between the two groups. For that reason, we adopted the accelerating features of the auditory stimulus used in the previous works.

Assaneo, M. F., Rimmele, J. M., Orpella, J., Ripollés, P., de Diego-Balaguer, R., & Poeppel, D. (2019). The Lateralization of Speech-Brain Coupling Is Differentially Modulated by Intrinsic Auditory and Top-Down Mechanisms. Frontiers in Integrative Neuroscience, 13. <https://doi.org/10.3389/fnint.2019.00028>

Assaneo, M. F., Rimmele, J. M., Sanz Perl, Y., & Poeppel, D. (2021). Speaking rhythmically can shape hearing. Nature Human Behaviour, 5(1), Article 1. <https://doi.org/10.1038/s41562-020-00962-0>

Assaneo, M. F., Ripollés, P., Orpella, J., Lin, W. M., de Diego-Balaguer, R., & Poeppel, D. (2019). Spontaneous synchronization to speech reveals neural mechanisms facilitating language learning. Nature Neuroscience, 22(4), 627–632. <https://doi.org/10.1038/s41593-019-0353-z>

Assaneo, M. F., Ripollés, P., Tichenor, S. E., Yaruss, J. S., & Jackson, E. S. (2022). The Relationship Between Auditory-Motor Integration, Interoceptive Awareness, and Self-Reported Stuttering Severity. Frontiers in Integrative Neuroscience, 16. <https://www.frontiersin.org/article/10.3389/fnint.2022.869571>

Lizcano-Cortés, F., Gómez-Varela, I., Mares, C., Wallisch, P., Orpella, J., Poeppel, D., Ripollés, P., & Assaneo, M. F. (2022). Speech-to-Speech Synchronization protocol to

classify human participants as high or low auditory-motor synchronizers. STAR Protocols.

Orpella, J., Assaneo, M. F., Ripollés, P., Noejovich, L., López-Barroso, D., Diego-Balaguer, R. de, & Poeppel, D. (2022). Differential activation of a frontoparietal network explains population-level differences in statistical learning from speech. PLOS Biology, 20(7), e3001712. <https://doi.org/10.1371/journal.pbio.3001712>

10) 366: Were the spoken stimuli in a male voice/female voice? Was the voice always the same?

Thanks for pointing this out. We included this information in this new version of the manuscript:

“Syllables were pseudo-randomly concatenated (the same syllable was never repeated consecutively) and the audio file was synthesized using the MBROLA text-to-speech synthesizer⁵³ with the Spanish Male Voice “es2” at 16 kHz.”

11) 370: what are the duration, and the rise and fall times of the tones?

We apologize for the poor description. The manuscript now reads:

“A train of 1 kHz tones was synthesized using MATLAB. Each tone was modulated with a triangular function with a fall time of 150 ms.”

12) 370-378: How was the tone frequency range determined? Were they based on previous works?

There is no consensus on one tone frequency for the finger tapping experiment (being the classic auditory-motor synchronization paradigm). Here, we chose 1 kHz for having a high cochlear response and for being within the range of natural environmental sounds (car horn ~500 Hz, bell ~1200 Hz, dog barking ~1500 Hz, vowels first and second formants 400-900 Hz 500-2500 Hz, respectively). However, we do not predict a modification in our pattern of results for a different tone frequency.

13) 374: Please describe the sine function to facilitate replication.

Again, we apologize for the poor description. The manuscript now reads:

“Tones were modulated by a sine function, so they had a soft onset and offset, and the same accelerating features of the previous stimuli were kept. The sine function was set to: $\sin(2\pi f t)$, with: $f=4.3$ if $t<10$, $f=4.4$ if $10<t<20$, $f=4.5$ if $20<t<30$, $f=4.6$ if $30<t<40$ and $f=4.7$ if $40<t$. A 50 second long train of these pseudo-randomly repeated tones (i.e., the same tone was never repeated consecutively) was synthesized using MATLAB.”

14) 382: How was it determined to use the 'tah' effector, as opposed to 'dah' or 'bah', for example?

As exposed in the answer to question 9 the present study builds on previous works of our team showing a bimodal distribution in the general population while assessing the speech-to-speech synchrony, all those studies used the effector "tah".

15) 364: So, seven total runs for each stimuli? Please clearly indicate the total number of runs for each condition/stimulus type.

We are not sure if we understand this comment correctly. We copied here the referred lines

362 All stimuli consisted of a rhythmic sequence of different acoustic units. The rhythmic structure
363 was constant across stimuli: the presentation rate of the acoustic units was progressively
364 increased from 4.3 to 4.7 units/sec, using steps of 0.1 units/sec; each rate was kept constant for
365 10 sec, so each stimulus lasted 50 sec. Four different stimuli were used throughout the study.

In the line pointed by the Reviewer we describe the general properties of the acoustic stimuli. Since different subjects completed different experiments with different stimulus-effector combinations, we explicit how many synchronization tasks completed each participant in the Experimental procedures section.

16) 408: it is stated the stimuli are presented at a mean of 70 db, yet earlier it is stated that subjects adjusted volume? Please clarify

We thank the Reviewer for highlighting this mistake. As correctly pointed by the Reviewer participants adjusted the volume and so stimuli were not presented at 70db. We apologize for the mistake, and we amended it in this new version of the manuscript, that now reads:

"All audio stimuli were presented binaurally at a variable sound pressure adjusted by the participant (the maximal volume reached by the used device and stimuli was 100 dB), via ETYMOTIC ER1 headphones, attached to ER1-14A earplugs."

17) How were effector onsets determined? By algorithm? By hand? What software was used? Related, how was alignment determined between stimuli timing & effector timing?

For the reasons stated in the answer to the Reviewer's suggestion (at the very beginning of the revision of this Reviewer) effectors' onsets were not determined. As stated in the Methods section the synchronization has been

estimated as the phase locking value between the envelope of the produced sounds. Regarding the phase lag between signals, it has been estimated by taking the phase evolution of the Hilbert transform of each envelope and taking the mean value for the phase difference. We would be happy to clarify this procedure further if the Reviewer considers it necessarily.

REVIEWERS' COMMENTS:

Reviewer #1 (Remarks to the Author):

The authors addressed all my concerns. I suggest publication.

Reviewer #2 (Remarks to the Author):

The authors have successfully addressed all of my concerns, and I have no further comments. Nice work!